# Decreased accuracy of erythrocyte sedimentation rate in diagnosing osteomyelitis in diabetic foot infection patients with severe renal impairment: A retrospective cross-sectional study

**Xin Chen**[1,2], **Yiting Shen**[1], **Yuying Wang**[1,2], **Yang Li**[2,3], **Shuyu Guo**[1], **Yue Liang**[1,2], **Xuanyu Wang**[1], **Siyuan Zhou**[1,2], **Xiaojie Hu**[1,2], **Kaiwen Ma**[1,2], **Rui Tian**[1,2], **Wenting Fei**[1,2], **Yuqin Sheng**[1,2], **Hengjie Cao**[1,2], **Huafa Que**[1] *

1 Department of Surgery of Traditional Chinese Medicine, Longhua Hospital Shanghai University of Traditional Chinese Medicine, Shanghai, China, 2 Longhua Medical College, Shanghai University of Traditional Chinese Medicine, Shanghai, China, 3 Department of Thoracic Surgery, Longhua Hospital Shanghai University of Traditional Chinese Medicine, Shanghai, China

☯ These authors contributed equally to this work.

* huafaque@126.com

**Data Availability Statement:** All relevant data are within the paper and its Supporting Information files.

## Abstract

### Background

Rapid diagnosis and treatment of diabetic foot osteomyelitis (DFO) could reduce the risk of amputation and death in patients with diabetic foot infection (DFI). Erythrocyte sedimentation rate (ESR) is considered the most useful serum inflammatory marker for the diagnosis of DFO. However, whether severe renal impairment (SRI) affects its diagnostic accuracy has not been reported previously.

### Objective

To investigate the accuracy of ESR in diagnosing DFO in DFI patients with and without SRI.

### Methods

This was a retrospective cross-sectional study. From the inpatient electronic medical record system, the investigators extracted demographic information, diagnostic information, and laboratory test results of patients with DFI who had been hospitalized in Longhua Hospital from January 1, 2016 to September 30, 2021. Logistic regression was performed to analyze the interaction between ESR and SRI with adjustment for potential confounders. The area under the curve (AUC), cutoff point, sensitivity, specificity, prevalence, positive predictive value (PPV), negative predictive value (NPV), positive likelihood ratio (LR+), and negative likelihood ratio (LR−) were analyzed by receiver operating characteristic (ROC) curve analysis and VassarStats.

**Funding:** HQ received research grant from the Shanghai Key Clinical Specialty (LH 02.91.002), from the Shanghai Municipal Health Commission. The funders had no role in study design, data collection and analysis, decision to publish, or preparation of the manuscript.

**Competing interests:** The authors have declared that no competing interests exist.

**Abbreviations:** AUC, area under the curve; BMI, body mass index; CI, confidence interval; DFI, diabetic foot infection; DFO, diabetic foot osteomyelitis; DPN, diabetic peripheral neuropathy; eGFR, estimated glomerular filtration rate; ESR, erythrocyte sedimentation rate; $HbA_{1c}$, glycosylated hemoglobin; hs-CRP, high-sensitivity C-reactive protein; IDSA, Infectious Diseases Society of America; IQR, interquartile range; IWGDF, International Working Group on the Diabetic Foot; LR+, positive likelihood ratio; LR−, negative likelihood ratio; MRI, magnetic resonance imaging; NPV, negative predictive value; OR, odds ratio; PAD, peripheral artery disease; PPV, positive predictive value; ROC, receiver operating characteristic; SRI, severe renal impairment; WBC, white blood cell.

## Results

A total of 364 DFI patients were included in the analysis. The logistic regression analysis results showed that elevated ESR increased the probability of diagnosing DFO (adjusted odds ratio [OR], 2.40; 95% confidence interval [CI], 1.75–3.28; adjusted $P < 0.001$); SRI was not associated with the diagnosis of DFO (adjusted OR, 3.20; 95% CI, 0.40–25.32; adjusted $P = 0.271$), but it had an obstructive effect on the diagnosis of DFO by ESR (adjusted OR, 0.48; 95% CI, 0.23–0.99; adjusted $P = 0.048$). ROC analysis in DFI patients without SRI revealed that the AUC of ESR to diagnose DFO was 0.76 (95% CI, 0.71–0.81), with the cutoff value of 45 mm/h (sensitivity, 67.8%; specificity, 78.0%; prevalence, 44.7%; PPV, 71.3%; NPV, 75.0%; LR+, 3.08; LR−, 0.41). In contrast, in patients with SRI, the AUC of ESR to diagnose DFO was 0.57 (95% CI, 0.40–0.75), with the cutoff value of 42 mm/h (sensitivity, 95.0%; specificity, 29.2%; prevalence, 45.5%; PPV, 52.8%; NPV, 87.5%; LR+, 1.34; LR−, 0.17).

## Conclusions

The accuracy of ESR in diagnosing DFO in DFI patients with SRI is reduced, and it may not have clinical diagnostic value in these patients.

## Introduction

Diabetic foot infection (DFI) is mostly caused by skin injury on a foot of a diabetic patient; it ranges from a superficial ulcer to a destruction of subcutaneous tissue, tendon, joint, and bone, which may eventually lead to amputation or death. Approximately 9.1 million to 26.1 million patients with diabetes develop foot ulcers each year [1], among which infected ulcers comprise about 41% [2, 3]. In recent years, the number of patients with diabetes admitted to the hospital due to infection has increased significantly [4]. Moderate and severe infections account for 47.4% of DFIs [5], and 23.5%–37.9% of patients with DFI have multiple bacterial infections [6, 7]. Patients with a history of ulcers, ulcers that have not healed for more than three months, and deep ulcers are more likely to develop infections [3]. Furthermore, infection significantly prolongs ulcer healing time and hospital stay, and it increases hospitalization costs [4]. The one-year healing rate of infected foot ulcers is only 45.5%; 9.6% of them recur later, and the lower extremity amputation rate is as high as 17.4% [8]. Diabetic foot patients with major amputations are more susceptible to medical complications and death [9]. Chronic open wounds lead to an increased risk of systemic infection and death within two years [10]; therefore, DFI is a huge health threat and causes a heavy socioeconomic burden.

Diabetic foot osteomyelitis (DFO) is a serious condition involving bones in patients with DFI. Almost 20% of patients with DFI develop osteomyelitis [11]. Diabetes patients with foot puncture injuries are nine times more likely to have osteomyelitis than non-diabetes patients and 14 times more likely to undergo amputation [12]. The amputation rate and mortality rate of DFO patients are 66.6% and 37.6% [13, 14], respectively, which are much higher than those of non-osteomyelitis patients with diabetes. The current gold standard for diagnosing DFO is bone biopsy, but in view of its invasiveness, it is commonly used only as the final diagnostic method when other approaches fail to make a clear diagnosis in clinical practice [15]. The guidelines of the International Working Group on the Diabetic Foot (IWGDF) recommend using a combination of probe-to-bone test, serum inflammatory markers, and plain

radiography for the initial diagnosis of DFO [15]. Among the serum inflammatory markers, erythrocyte sedimentation rate (ESR) is particularly important.

Patients with diabetes are seven times more likely to suffer from kidney disease [12], and type 2 diabetes has become the main cause of end-stage renal disease [16]. A recent study has shown that patients with diabetic nephropathy have higher ESR than those with non-diabetic nephropathy, and ESR is an independent risk factor for diabetic nephropathy [17]. Another study revealed that ESR was 49±26 mm/h in patients with chronic renal failure who did not receive dialysis, and it was as high as 60±33 mm/h in patients receiving hemodialysis [18]. Hence, with the decline of renal function, ESR increases in patients without active infection. In patients with current infection, differences in ESR elevation between patients with and without renal impairment have not been reported before.

Considering the high disability and mortality rate of DFO, its timely diagnosis is of paramount importance. In contrast to the invasiveness of bone biopsy, ESR is a timely and relatively practical diagnostic marker. However, there is not enough information as to whether renal function affects its diagnostic accuracy. Therefore, it is necessary to study the accuracy of ESR in diagnosing osteomyelitis in DFI patients with different levels of renal function.

## Methods

### Study design and ethical approval

This was a retrospective cross-sectional study and was reported in compliance with The Strengthening the Reporting of Observational Studies in Epidemiology statement (S1 File). All of the researchers were systematically trained before the study to fully understand the details of the study and their responsibilities. This study was reviewed and approved by the Institutional Review Board of Longhua Hospital Shanghai University of Traditional Chinese Medicine (2021LCSY115). The patients were contacted through the contact information stored in the electronic medical record system if they satisfied the criteria for inclusion. Investigators informed the patients that necessary information during a certain hospitalization would be used while ensuring that their personal privacy would not be leaked; the patients came to the hospital to sign the written informed consent form. If a patient did not want to come to the hospital to sign the form, he or she was informed about the details on the phone and his oral informed consent was obtained and recorded in the list. If a patient could not be contacted at all, in accordance with the *Declaration of Helsinki* from 2008 and the No. 11 Order of the National Health Commission of the People's Republic of China, after review and approval by the Institutional Review Board, an exemption of informed consent was applied. From the inpatient electronic medical record system, the investigators extracted the necessary information of patients with DFI who had been hospitalized in Longhua Hospital Shanghai University of Traditional Chinese Medicine from January 1, 2016 to September 30, 2021. The investigators accessed medical records, screened included patients, and contacted patients to obtain informed consent from October 29, 2021 to November 8, 2021. The statistical analysis was completed on November 12, 2021.

### Diagnostic criteria

The diagnosis of DFI and DFO was based on the 2015 IWGDF guidelines [19].

The diagnosis of DFI was based on local or systemic inflammatory symptoms and clinical signs, and its severity was assessed using the Infectious Diseases Society of America (IDSA)/ IWGDF classification scheme. Erythema, swelling, induration, pain, tenderness, or pus on the foot indicated a local infection. When systemic inflammatory response syndrome occurred,

the infection had affected the whole body. According to the IDSA/IWGDF classification scheme, it was classified as mild, moderate, and severe infection.

The diagnosis of DFO was based on a comprehensive evaluation of positive probe-to-bone test, and abnormal plain radiography findings. If the diagnosis was not clear, foot magnetic resonance imaging (MRI) was further implemented. If the diagnosis of DFO was not convincing, bone biopsy was used to finally confirm the diagnosis. The probe-to-bone test were conducted within 12 hours of admission, and the plain radiography was completed within 24 hours. If necessary, MRI was completed within five days, and bone biopsy was performed within seven days.

The criterion of severe renal impairment (SRI) was based on the guidelines developed by the National Kidney Foundation [20]. SRI is characterized by severe reduction in estimated glomerular filtration rate (eGFR) (eGFR $< 30$ mL/min/1.73 m$^2$).

## Inclusion and exclusion criteria

Patients hospitalized in Longhua Hospital for the first time and meeting the DFI diagnostic criteria were included. Patients with rheumatic diseases, inflammatory bowel disease, and other inflammatory diseases that can cause elevated ESR were excluded. In addition, patients in whom the probe-to-bone test, laboratory examination (within 24 hours of admission), plain radiography or MRI, and bone biopsy were not completed within the prescribed time limit were excluded. Finally, incomplete medical record without sufficient data was also ruled out.

## Data collection

The investigators extracted the required data from the inpatient electronic medical record system, including demographic information, diagnostic information, and laboratory tests results. Demographic information included the patient's age, gender, body mass index (BMI), diabetes duration, and infection duration. The infection duration referred to the time from the earliest occurrence of any local infection symptoms described in the DFI diagnostic criteria to the hospital admission. The diagnostic information included the signs of severe infection (IDSA/IWGDF classification scheme grade 4), presence of DFO, peripheral artery disease (PAD), and diabetic peripheral neuropathy (DPN). Laboratory tests included white blood cell (WBC) count, high-sensitivity C-reactive protein (hs-CRP) level, ESR, creatinine level, and glycosylated hemoglobin (HbA$_{1c}$) level. The laboratory tests were carried out by a nurse taking fasting blood for examination. ESR was measured using an Automated ESR Analyzer Test 1 (Italy ALIFAX Corp.). Creatinine was measured with a Beckman Coulter analyzer AU5800 using a creatinine enzymatic kit (Beckman Coulter Ireland Inc.) by the creatine oxidase method. The eGFR was calculated using the CKD-EPI formula based on creatinine [21]. Two investigators were responsible for information extraction over a specific period. If the information was inconsistent, they checked it, and if necessary, consulted with a third senior researcher for arbitration.

## Statistical analysis

The collected data were analyzed by independent statisticians. Continuous data with normal distribution were presented as mean (standard deviation); non-normal distribution data were presented as median (interquartile range [IQR]). Categorical data were reported as N (%). The continuous data with normal distribution were compared by one-way analysis of variance, and the continuous data not conforming to normal distribution were compared by Kruskal–Wallis *H* test. Categorical variables were compared using Pearson chi-square test or Fisher's exact probability test if expected numbers were small. The interaction between ESR and SRI was

analyzed by logistic regression. Variables that showed marginal statistical difference between patients with and without SRI were considered as a potential confounders and were included in the logistic regression. The area under the curve (AUC), cutoff point, sensitivity, and specificity were analyzed by receiver operating characteristic (ROC) curve. The Youden index was used to determine the optimal cutoff point. It was calculated as follows: Youden index = sensitivity + specificity − 1. Prevalence, positive predictive value (PPV), negative predictive value (NPV), positive likelihood ratio (LR+), and negative likelihood ratio (LR−) at optimal cutoff point were calculated by Clinical Calculator 1 of VassarStats website (http://vassarstats.net/clin1.html). A two-tailed $P$ value ≤ 0.05 was considered significant. The data were analyzed by SPSS version 22.0 for Windows (SPSS Inc., Chicago, IL, USA).

## Results

### Clinical characteristics of patients with DFI

Of 1066 screened medical records, 373 patients met the inclusion criteria, and nine of them were excluded. Of the nine excluded patients, two patients with rheumatoid arthritis, and seven patients lacked necessary data in their medical records. In these incomplete medical records, three patients had no laboratory test results because they refused to take fasting blood, and no definite diagnosis was made in four patients with suspected DFO. One patient requested to be discharged prior to MRI. One patient experienced an acute cerebrovascular event, hence a planned bone biopsy was abandoned. Two patients refused bone biopsy. The flow chart of this study is shown in Fig 1. A total of 364 DFI patients with complete data were included in the analysis, including 163 patients with osteomyelitis and 201 patients without osteomyelitis. The study population was predominantly normal renal function or mild to moderately impaired (87.9%) and female (66.8%); mean age was 70 years (range, 36–97 years). Of the 44 patients with SRI, 19 patients received hemodialysis, whose eGFR all less than 15 mL/min/1.73 m$^2$, including 10 patients with DFO and nine patients without DFO. Differences in gender, infection duration, signs of severe infection, eGFR, ESR, WBC count, hs-CRP level, and HbA$_{1c}$ level were statistically significant among the four groups ($P < 0.05$). No significant differences were found in age, BMI, diabetes duration, presence of PAD, and DPN between groups ($P > 0.05$). Patient characteristics are summarized in Table 1.

### Analysis of the interaction between ESR and SRI

Variables (gender, diabetes duration, PAD, hs-CRP, and HbA$_{1c}$) with a univariate $P \leq 0.1$ were considered as potential confounders (Table 2). The variables of interest, including ESR, SRI, and interaction between ESR and SRI, were included in the binary logistic regression equation, and adjusted by potential confounders. The continuous variables included in logistic regression analysis were converted into categorical variables to facilitate clinical interpretation (assignment rules: ESR: ≤ 30 mm/h, 1, > 30 to 60 mm/h, 2, > 60 to 90 mm/h, 3, and > 90 mm/h, 4; diabetes duration: ≤ 10 years, 1, > 10 to 20 years, 2, > 20 to 30 years, 3, > 30 years, 4; hs-CRP: ≤ 30.00 mg/L, 1, > 30.00 to 60.00 mg/L, 2, > 60.00 to 90.00 mg/L, 3, > 90.00 mg/L, 4; HbA$_{1c}$: ≤ 7.0%, 1, > 7.0 to 9.0%, 2, > 9.0 to 11.0%, 3, > 11.0%, 4). The analysis results showed that every 30 mm/h increase in ESR increased the risk of diagnosing DFO (adjusted odds ratio [OR], 2.40; 95% confidence interval [CI], 1.75–3.28; adjusted $P < 0.001$); SRI had no effect on the diagnosis of DFO (adjusted OR, 3.20; 95% CI, 0.40–25.32; adjusted $P = 0.271$), but it had an obstructive effect on the diagnosis of DFO by ESR (adjusted OR, 0.48; 95% CI, 0.23–0.99; adjusted $P = 0.048$). The main results from logistic regression analysis are shown in Table 3.

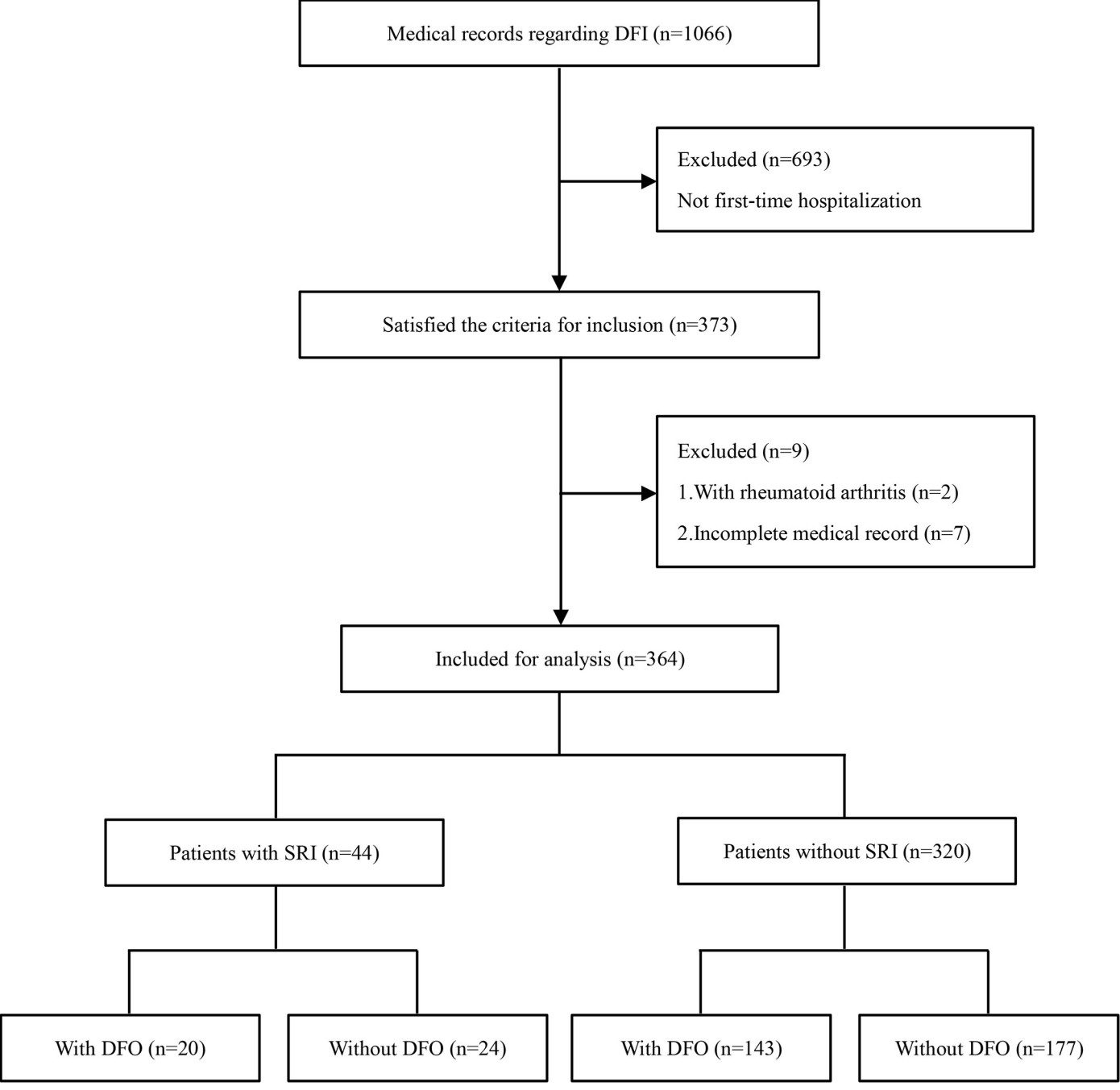

**Fig 1. The flow chart of the study.** DFI, diabetic foot infection; DFO, diabetic foot osteomyelitis; SRI, severe renal impairment.

### Accuracy of ESR in diagnosing DFO in patients with and without SRI

Of the 364 patients with DFI, 44 patients (mean age, 69 years, range, 36–95 years; 47.7% male) met SRI standard. The average age of 320 patients without SRI was 70 years (range, 41–97 years), 31.3% were male. The median ESR in the patients with SRI was 70 mm/h (IQR, 47–90 mm/h); among patients without SRI the median ESR was 40 mm/h (IQR, 23–63 mm/h). The Box–Whisker plot shows a significant difference ($P < 0.001$; Kruskal–Wallis $H$ test) in ESR

**Table 1. Characteristics of patients with DFI.**

| | SRI (n = 44) | | No SRI (n = 320) | | P value |
| --- | --- | --- | --- | --- | --- |
| | DFO (n = 20) | No DFO (n = 24) | DFO (n = 143) | No DFO (n = 177) | |
| **Age (years)** | 67 (8) | 71 (12) | 68 (12) | 71 (11) | 0.151[c] |
| **Gender**[a] | 11 (55.0%) | 10 (41.7%) | 55 (38.5%) | 45 (25.4%) | 0.009[d] |
| **BMI (kg/m²)** | 23.89 (2.88) | 22.36 (2.52) | 23.69 (3.14) | 23.67 (3.36) | 0.268[c] |
| **Diabetes duration (years)** | 19 (16, 30) | 18 (11, 23) | 16 (10, 21) | 16 (10, 21) | 0.124[e] |
| **Infection duration (weeks)** | 8 (3, 25) | 4 (2, 12) | 5 (3, 12) | 4 (2, 11) | 0.041[e] |
| **Signs of severe infection**[b] | 3 (15.0%) | 8 (33.3%) | 37 (25.9%) | 18 (10.2%) | < 0.001[f] |
| **PAD** | 18 (90.0%) | 22 (91.7%) | 116 (81.1%) | 140 (79.1%) | 0.416[f] |
| **DPN** | 9 (45.0%) | 11 (45.8%) | 86 (60.1%) | 100 (56.5%) | 0.398[d] |
| **eGFR (mL/min/1.73 m²)** | 14.80 (7.62, 23.85) | 18.37 (9.65, 26.12) | 91.66 (68.20, 105.55) | 85.19 (60.88, 98.54) | < 0.001[e] |
| **ESR (mm/h)** | 73 (53, 89) | 61 (39, 90) | 59 (39, 71) | 28 (16, 44) | < 0.001[e] |
| **WBC (×10⁹/L)** | 8.44 (6.72, 11.38) | 8.83 (6.33, 13.41) | 8.55 (6.62, 12.77) | 7.36 (5.94, 9.35) | 0.001[e] |
| **hs-CRP (mg/L)** | 39.33 (7.72, 113.27) | 27.35 (4.51, 73.78) | 24.02 (6.18, 68.7) | 3.09 (0.62, 16.12) | < 0.001[e] |
| **HbA₁c (%)** | 7.1 (6.5, 8.6) | 8.0 (6.7, 8.6) | 9.2 (7.4, 10.7) | 8.3 (7.1, 9.8) | < 0.001[e] |

Age and BMI are expressed as mean (standard deviation); diabetes duration, infection duration, eGFR, ESR, WBC, hs-CRP, and HbA₁c are expressed as median (IQR); gender, signs of severe infection, PAD, and DPN are expressed as N (%).

[a]N of male;

[b]IDSA/IWGDF classification scheme grade 4;

[c]one-way analysis of variance;

[d]Pearson chi-square test;

[e]Kruskal–Wallis $H$ test;

[f]Fisher's exact probability test.

**Table 2. Univariate analysis between patients with and without SRI.**

| | SRI (n = 44) | No SRI (n = 320) | P value |
| --- | --- | --- | --- |
| **Age (years)** | 69 (10) | 70 (11) | 0.907[c] |
| **Gender**[a] | 21 (47.7%) | 100 (31.3%) | 0.030[d] |
| **BMI (kg/m²)** | 23.05 (2.76) | 23.68 (3.26) | 0.226[c] |
| **Diabetes duration (years)** | 19 (13, 26) | 16 (10, 21) | 0.046[e] |
| **Infection duration (weeks)** | 5 (3, 15) | 4 (2, 12) | 0.184[e] |
| **Signs of severe infection**[b] | 11 (25.0%) | 55 (17.2%) | 0.207[d] |
| **PAD** | 40 (90.9%) | 256 (80.0%) | 0.082[d] |
| **DPN** | 20 (83.3%) | 186 (58.1%) | 0.112[d] |
| **WBC (×10⁹/L)** | 8.44 (6.63, 12.46) | 7.83 (6.20, 10.53) | 0.150[e] |
| **hs-CRP (mg/L)** | 29.88 (6.29, 97.56) | 8.41 (1.29, 37.39) | 0.001[e] |
| **HbA₁c (%)** | 7.7 (6.6, 8.6) | 8.7 (7.2, 10.3) | 0.001[e] |

Age and BMI are expressed as mean (standard deviation); diabetes duration, infection duration, WBC, hs-CRP, and HbA₁c are expressed as median (IQR); gender, signs of severe infection, PAD, and DPN are expressed as N (%).

[a]N of male;

[b]IDSA/IWGDF classification scheme grade 4;

[c]one-way analysis of variance;

[d]Pearson chi-square test;

[e]Kruskal–Wallis $H$ test.

**Table 3. Analysis of the interaction between ESR and SRI in diagnosis of DFO.**

|  | Unadjusted | | | Adjusted | | |
|---|---|---|---|---|---|---|
|  | OR value | 95% CI for OR value | *P* value | OR value | 95% CI for OR value | *P* value |
| ESR | 2.71 | 2.03–3.62 | < 0.001 | 2.40 | 1.75–3.28 | < 0.001 |
| SRI | 4.00 | 0.53–30.15 | 0.179 | 3.20 | 0.40–25.32 | 0.271 |
| ESR×SRI[a] | 0.46 | 0.22–0.95 | 0.036 | 0.48 | 0.23–0.99 | 0.048 |

Adjusting for gender, diabetes duration, PAD, hs-CRP, and HbA$_{1c}$.
[a]Interaction between ESR and SRI.

between patients with and without SRI (Fig 2). According to the ROC analysis, the AUC of ESR in diagnosis of DFO was 0.74 (95% CI, 0.68–0.79, $P < 0.001$), with the cutoff value of 45 mm/h, sensitivity of 70.6% (95% CI, 62.8%–77.3%), specificity of 72.1% (95% CI, 65.3%–78.1%), prevalence of 44.8% (95% CI, 39.6%–50.1%), PPV of 67.3% (95% CI, 59.6%–74.1%), NPV of 75.1% (95% CI, 68.3%–80.9%), LR+ of 2.53 (95% CI, 1.98–3.23), and LR–of 0.41 (95% CI, 0.32–0.52). In patients without SRI, the AUC of ESR to diagnose DFO was 0.76 (95% CI, 0.71–0.81, $P < 0.001$), with the cutoff value of 45 mm/h, sensitivity of 67.8% (95% CI, 59.4%–75.3%), specificity of 78.0% (95% CI, 71.0%–83.7%), prevalence of 44.7% (95% CI, 39.2%–50.3%), PPV of 71.3% (95% CI, 62.8%–78.6%), NPV of 75.0% (95% CI, 68.0%–80.9%), LR+ of 3.08 (95% CI, 2.28–4.15), and LR–of 0.41 (95% CI, 0.32–0.53). In contrast, in patients with SRI, the AUC of ESR in diagnosis of DFO was 0.57 (95% CI, 0.40–0.75, $P = 0.409$), with the cutoff value of 42 mm/h, sensitivity of 95.0% (95% CI, 73.1%–99.7%), specificity of 29.2% (95% CI, 13.4%–51.2%), prevalence of 45.5% (95% CI, 30.7%–61.0%), PPV of 52.8% (95% CI, 35.7%–69.2%), NPV of 87.5% (95% CI, 46.7%–99.3%), LR+ of 1.34 (95% CI, 1.02–1.77), and LR–of 0.17 (95% CI, 0.02–1.41). The ROC analyses of all of the patients and the two subgroups are shown in Fig 3. The sensitivity, specificity, PPV, NPV, LR+, and LR–with their 95% CIs of all of the patients and the two subgroups are shown in Fig 4.

## Discussion

Studies in recent decades have shown that ESR is a reliable serum inflammatory marker for the diagnosis of DFO. The results of a prospective study have revealed that ESR > 67 mm/h has the optimal sensitivity (84%) and specificity (75%) for the diagnosis of DFO [22]. Another retrospective study involving 353 patients showed that when ESR was > 60 mm/h, the sensitivity of diagnosing DFO was 74% and the specificity was 56% [23]. A recent study has shown that the AUC of ESR to diagnose DFO was 0.70, with the cutoff value of 49 mm/h, sensitivity of 74.6%, and specificity of 57.7% [24]. A study in a Chinese population showed that the AUC of ESR for the diagnosis of DFO was 0.832, with the cutoff value of 43 mm/h, sensitivity of 82.9%, and specificity of 70.5% [25]. ESR has been shown to increase in patients with renal dysfunction [26, 27]. In a retrospective study of 200 patients, the average ESR of patients with chronic kidney disease stage 3 or stage 4, hemodialysis, and peritoneal dialysis was 42.71 ±27.60 mm/h, 45.26±29.03 mm/h, and 39.92±28.24 mm/h, respectively; in contrast, the average ESR of kidney transplant patients was 25.95±21.51 mm/h [28]. Another study investigating the changes in ESR in stable hemodialysis patients revealed that the average ESR before dialysis was as high as 49.8±28.5 mm/h, and that it increased to 55.6±30.4 mm/h after dialysis [29]. In patients with diabetic nephropathy, ESR increased to 52.02±27.71 mm/h [30]. Hence, it is clear that with the decline in renal function, ESR increases in patients without recent infection; however, its impact on the diagnosis of DFO has not been investigated previously.

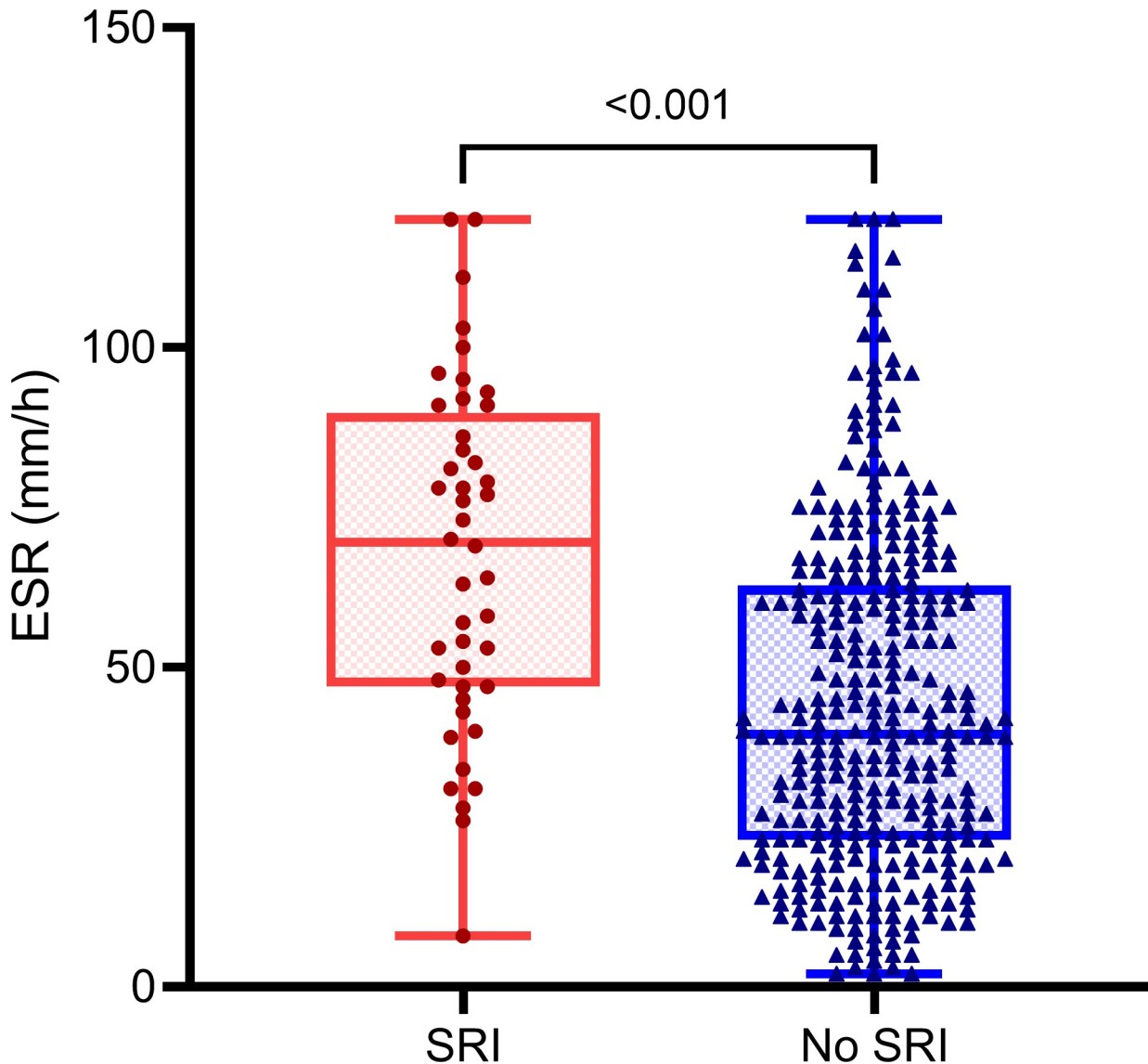

**Fig 2. Box–Whisker plot of the difference in ESR between DFI patients with and without SRI.** The graph presents the medians, quartiles, and range of ESR in DFI patients with and without SRI. The red dots and blue triangles show all individual data points of two groups of patients. The median ESR of the 44 patients with SRI was 70mm/h (IQR, 47–90 mm/h); and the median ESR of the 320 patients without SRI was 40mm/h (IQR, 23–63 mm/h). There was a statistically significant difference between patients with and without SRI ($P < 0.001$; Kruskal–Wallis $H$ test). DFI, diabetic foot infection; ESR, erythrocyte sedimentation rate; IQR, interquartile range; SRI, severe renal impairment.

There are three main findings from our research. First, there was an interaction between ESR and SRI in diagnosis of DFO. Logistic regression analysis showed that elevated ESR was associated with a higher probability of DFO; SRI was not directly associated with the diagnosis of DFO, but it hindered the diagnosis of DFO by ESR. Second, the diagnostic accuracy of ESR in DFI patients with SRI is lower, so that it may have no clinical reference value. In DFI patients without SRI, there was a significant difference in ESR between patients with osteomyelitis and those without osteomyelitis. In contrast, in DFI patients with SRI, no statistical difference in ESR was observed between the two subgroups. When renal function is normal or mild to moderate impairment, the AUC of ESR to diagnose DFO is 0.76, which has a reasonable diagnostic value, but when renal impairment severely,

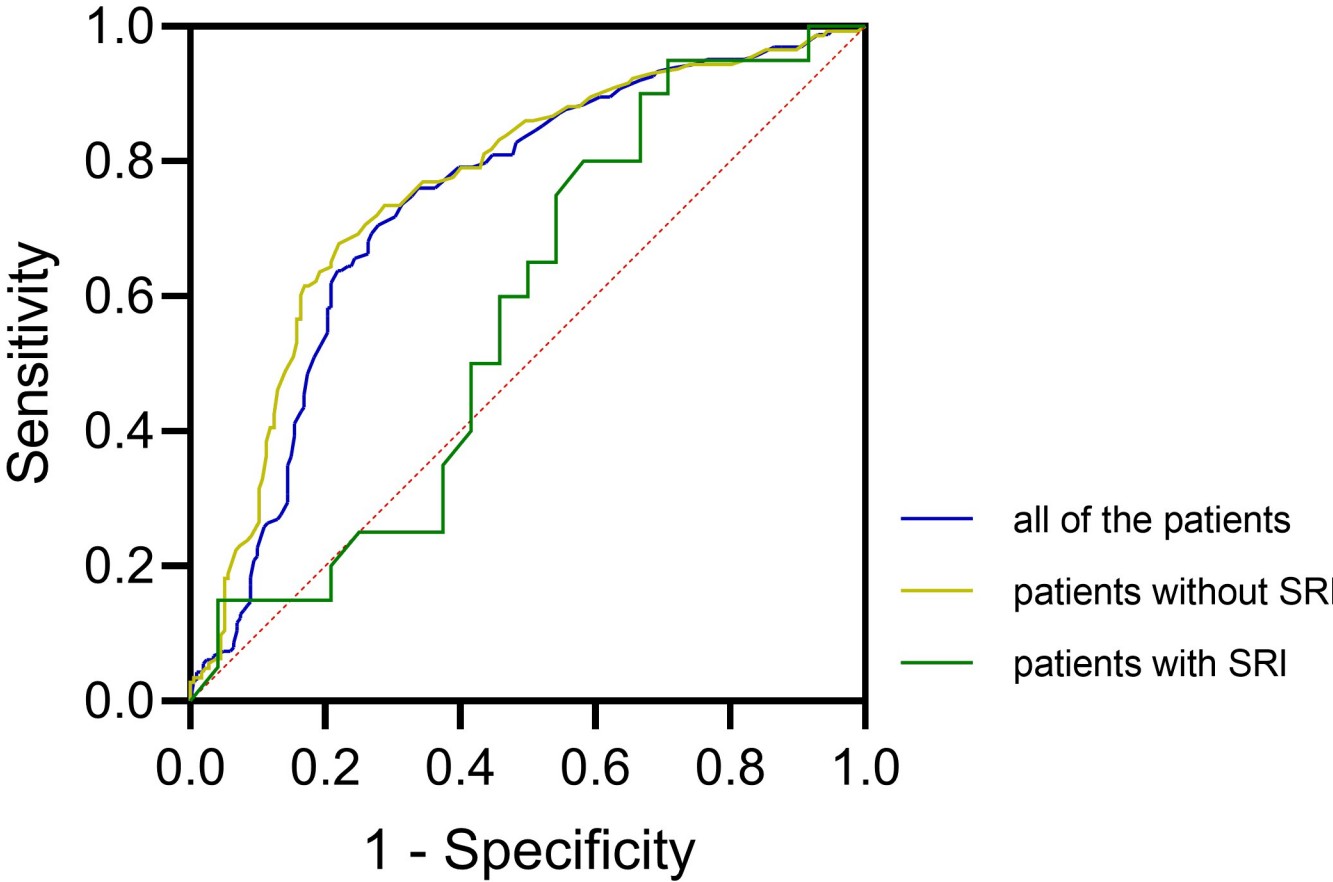

**Fig 3. ROC analysis for all of the patients and the two subgroups.** The graph illustrates ROC analysis results for all of the patients (blue solid line), patients without SRI (yellow solid line), and patients with SRI (green solid line). It can be seen that all of the patients and patients without SRI share similar curves, but the curve of patients with SRI is far away from them and close to the expected line representing the performance of random guess (diagonal red dashed line). Moreover, the AUC of patients with SRI is distinctly smaller than the other two groups (patients with SRI, 0.57; patients without SRI, 0.76; all of the patients, 0.74). These indicate ESR has a limited value on predicting DFO in patients with SRI. AUC, area under the curve; DFO, diabetic foot osteomyelitis; ESR, erythrocyte sedimentation rate; ROC, receiver operating characteristic; SRI, severe renal impairment.

the AUC drops to 0.57, which means that ESR has a limited value on predicting DFO. The specificity of ESR to diagnose DFO in patients without SRI is 78.0%, moreover, the PPV at about 45% prevalence and LR+ are 71.3% and 3.08, respectively. For patients with SRI, by contrast, the specificity of ESR in diagnosis of DFO dramatically slides to 29.2%, furthermore, the PPV drops to 52.8% at a similar prevalence, and LR+ also noticeably declines to 1.34. Significantly decreased specificity, PPV, and LR+ indicates that ESR has a poor clinical diagnostic value in patients with SRI. Third, although SRI showed a great interference with the diagnosis of DFO by ESR in our study, the proportion of such patients only accounts for 12.1% of the total sample, which has a small impact on the overall diagnosis accuracy. The AUC was similar in the entire sample and subgroup without SRI, with the same cutoff value of 45 mm/h; there was little difference in sensitivity, specificity, PPV, NPV, LR+, and LR–at the optimal cutoff point. This indicates that the previous research results are applicable to most situations. However, when it is estimated that the patient's renal function is severely reduced, it is necessary to be aware that ESR should not be used as a reference for diagnosing DFO, but the preliminary diagnosis should be made based on other methods, such as probe-to-bone test or plain radiography.

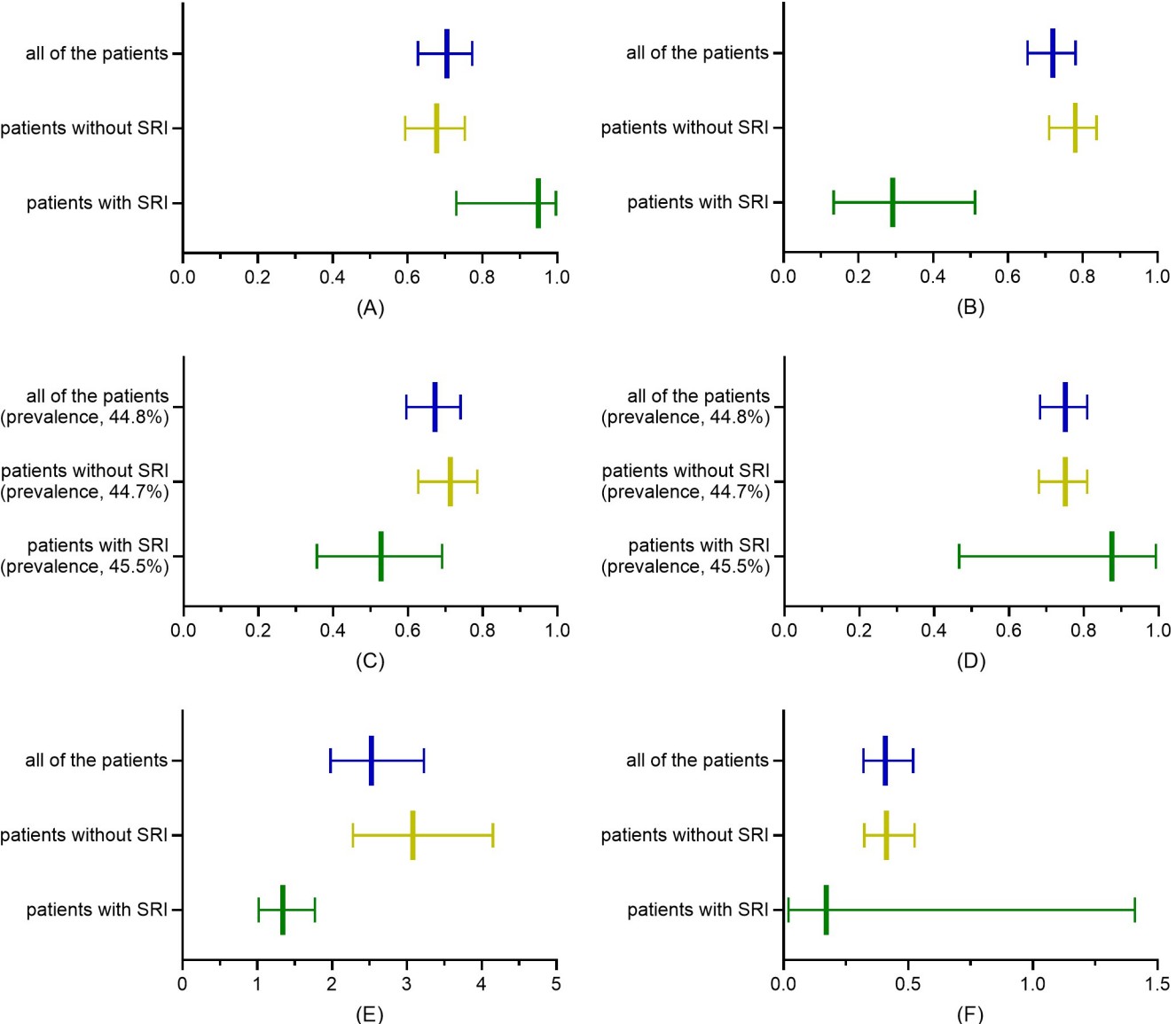

**Fig 4. Box–Whisker plot of sensitivity, specificity, PPV, NPV, LR+, and LR–for all of the patients and the two subgroups.** The panels summary the sensitivity (A), specificity (B), PPV (C), NPV (D), LR+ (E), and LR–(F) with their 95% CIs at optimal cutoff points for all of the patients (45 mm/h), patients without SRI (45 mm/h), and patients with SRI (42 mm/h), which were calculated by VassarStats. It is apparent that the sensitivity, specificity, PPV, NPV, LR +, and LR–between all of the patients and patients without SRI are generally similar, with widely overlapping 95% CIs. What is interesting is that, the specificity, PPV, and LR+ in patients with SRI are strikingly lower than the other two groups and 95% CIs do not widely overlap with them. These implies a declining predictive value of DFO by ESR in patients with SRI. CI, confidence interval; DFO, diabetic foot osteomyelitis; ESR, erythrocyte sedimentation rate; LR+, positive likelihood ratio; LR−, negative likelihood ratio; NPV, negative predictive value; PPV, positive predictive value; SRI, severe renal impairment.

There are some limitations in this study. First, this was a retrospective study, and all of the necessary data were extracted from an inpatient electronic medical record system. Second, although we included a large number of cases, the optimal sample size was not calculated due to the lack of relevant literature support. Third, most patients were diagnosed with DFO based on clinical symptoms, signs, and auxiliary examination results, and only a fraction of patients underwent bone biopsy. For clinical practice, current reference standard could guide diagnosis and treatment well. Nevertheless, for diagnostic accuracy test, it is possible that DFO is

misdiagnosed. Therefore, there is still a need to develop a reference standard that takes into account both diagnostic accuracy and clinical utility. Fourth, some studies have reported that ESR increased after dialysis compared with before dialysis in stable dialysis patients, but there was no related report in patients with active infection. We would observe the influence of blood collection times on ESR of infected patients, so as to consider blood collection time of dialysis patients as one of inclusion criteria. These conditions may have led to research bias. Therefore, a well-designed prospective study with *a priori* calculated sample size is needed to verify the reliability of our conclusions.

## Conclusion

The present study was designed to assess the accuracy of ESR to diagnose DFO in patients with SRI. The evidence gained here could help to understand the role of SRI in the diagnosis of DFO by ESR. We have presented that when DFI patients without SRI (eGFR $\geq$ 30 mL/min/1.73 m$^2$), the AUC, specificity, PPV, and LR+ were 0.76, 78.0%, 71.3%, and 3.08, respectively. When DFI patients with SRI (eGFR $<$ 30 mL/min/1.73 m$^2$), the AUC (0.57), specificity (29.2%), PPV (52.8%), and LR+ (1.34) of ESR in diagnosing DFO decreased significantly, so that it could not distinguish well whether there is osteomyelitis in DFI patients. Previous studies have focused on fast and early diagnosis of DFO based on ESR, but the results of our study emphasize that the use of ESR to diagnose DFO needs to consider SRI to make a reasonable judgment, and other methods should be used for preliminary diagnosis when renal function is severely reduced. Our study provided a deeper insight into the relationship between DFI and SRI, and contributed to the understanding of the diagnosis of DFO by ESR. But with regard to the retrospective research method, some important weaknesses need to be acknowledged. Further prospective investigation will be needed based on a sample size calculated by findings from current work.

## Supporting information

**S1 File. STROBE checklist.**
(DOCX)

**S1 Data. Minimal anonymized data set.**
(XLSX)

## Author Contributions

**Conceptualization:** Xin Chen, Huafa Que.

**Data curation:** Xin Chen, Yiting Shen, Yuying Wang, Yang Li, Shuyu Guo, Yue Liang, Xuanyu Wang, Siyuan Zhou, Xiaojie Hu, Kaiwen Ma, Rui Tian, Wenting Fei, Yuqin Sheng, Hengjie Cao, Huafa Que.

**Formal analysis:** Xin Chen, Yang Li.

**Funding acquisition:** Huafa Que.

**Investigation:** Yiting Shen, Yuying Wang, Shuyu Guo, Yue Liang, Xuanyu Wang, Siyuan Zhou, Xiaojie Hu, Kaiwen Ma, Rui Tian, Wenting Fei, Yuqin Sheng, Hengjie Cao.

**Methodology:** Xin Chen, Yang Li, Huafa Que.

**Project administration:** Huafa Que.

**Supervision:** Huafa Que.

**Writing – original draft:** Xin Chen, Yiting Shen, Yuying Wang.

**Writing – review & editing:** Xin Chen, Yiting Shen, Yuying Wang, Yang Li, Shuyu Guo, Yue Liang, Xuanyu Wang, Siyuan Zhou, Xiaojie Hu, Kaiwen Ma, Rui Tian, Wenting Fei, Yuqin Sheng, Hengjie Cao, Huafa Que.

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
