## [Decision Letter · Decision Letter 0]

10 Jan 2022

PONE-D-21-37187Decreased accuracy of erythrocyte sedimentation rate in diagnosing osteomyelitis in diabetic foot infection patients with severe renal impairment: a retrospective cross-sectional studyPLOS ONE

Dear Dr. Que,

Thank you for submitting your manuscript to PLOS ONE. After careful consideration, we feel that it has merit but does not fully meet PLOS ONE’s publication criteria as it currently stands. Therefore, we invite you to submit a revised version of the manuscript that addresses the points raised during the review process.

We look forward to receiving your revised manuscript.

Kind regards,

Kanhaiya Singh, Ph.D

Academic Editor

PLOS ONE

Journal Requirements:

Additional Editor Comments:

Although reviewers have found this study interesting, they have raised some significant concerns. Please address the novelty and study concerns as mentioned by Reviewer 1. Also, please discuss in details about the excluded subjects.

Reviewers' comments:

Reviewer's Responses to Questions

**Comments to the Author**

1. Is the manuscript technically sound, and do the data support the conclusions?

Reviewer #1: Partly

Reviewer #2: Yes

Reviewer #3: Partly

2. Has the statistical analysis been performed appropriately and rigorously? 

Reviewer #1: N/A

Reviewer #2: Yes

Reviewer #3: Yes

3. Have the authors made all data underlying the findings in their manuscript fully available?

Reviewer #1: No

Reviewer #2: Yes

Reviewer #3: Yes

4. Is the manuscript presented in an intelligible fashion and written in standard English?

Reviewer #1: Yes

Reviewer #2: Yes

Reviewer #3: Yes

5. Review Comments to the Author

Reviewer #1: Ref: PONE-D-21-37187

In the present article entitled “Decreased accuracy of erythrocyte sedimentation rate in diagnosing osteomyelitis in

diabetic foot infection patients with severe renal impairment: a retrospective cross-sectional

study” Que et al., have attempted to investigate the accuracy of erythrocyte sedimentation rate (ESR) in

diagnosing diabetic foot osteomyelitis (DFO) in diabetic foot infection (DFI) patients

with and without severe renal impairment (SRI). Although the manuscript contains extensive amount of work, I have several concerns about study design, data presentation and interpretation. Here are the specific comments.

Abstract: The starting paragraph is not scientifically developed. If there is already plethora of studies, why authors want to study this? There is no gap of knowledge is shown.

1. Inclusion criteria: The authors have studied patients with severe renal impairment: No mention has been made of patients receiving Hemodialysis/ peritoneal dialysis for the diabetic nephropathy. As noted in a previous study

by Alisomali MI et al (ref no 29): Post-dialysis the ESR was raised in most of the stable patients on regular HD and was significantly higher than the pre-dialysis ESR (by, on average, 5.8 mm/h).

2. Novelty concern and Study Design concerns: The same study (ref no 29) and several other studies (cited below) have mentioned that 'high ESR may be limited diagnostic utility in patients with Chronic kidney disease'. Making the entire diagnostic value in patients specifically having Diabetic foot infection and osteomyelitis redundant.

Barthon J, Graves J, Jens P, et al. The erythrocyte sedimentation rate in end-stage renal failure. Am J Kidney Dis 1987;10: 34-40. https://www.ncbi.nlm.nih.gov/pubmed/3605082

Shusterman N, Morrison G, Singer I. The erythrocyte sedimentation rate and chronic renal failure. Ann Intern Med 1986;105:801. http://annals.org/aim/fullarticle/700910

Arik N, Bedir A, Gunaydin M, et al. Do erythrocyte sedimentation rate and C-reactive protein levels have diagnostic usefulness in patients with renal failure? Nephron 2000;86:224. https://www.ncbi.nlm.nih.gov/pubmed/11015011

Warner DM, George CRP. Erythrocyte sedimentation rate and related factors in end-stage renal failure. Nephron 1991;57:248. https://www.karger.com/Article/PDF/186266

As the authors mention later in study limitations, the diagnosis of Diabetic foot osteomyelitis has not been established by bone biopsy in majority which is a gold standard.(line 251)

Fig 2 'The difference in ESR between DFI patients with and without SRI': Individual data points could be shown in the Box- Whisker plot to compare two variables. Also the legend does not mention the type of graph or analysis used (Kruksal - Wallis H test)

Fig 3: more could have been done with the present data if authors could find the Positive and negative predictive value using ROC curves and prevalence of disease in population.

To conclude ESR is inherently a very non-specific inflammatory marker to begin with which can be raised in diabetes mellitus and its several associated complications.

Best regards,

Reviewer #2: The role of ESR in making a diagnosis of osteomyelitis in diabetic foot ulcers is complimentary to other investigations including imaging and bone scan. This is an interesting study stating that the adjunctive value of ESR in diagnosing osteomyelitis diminishes in patients with severe renal impairment. ESR may therefore be used as a screening test for making a presumptive diagnosis of osteomyelitis in patients with diabetic foot infections without renal impairment

Reviewer #3: 1. In line number 71, authors mentioned there is a high chance of diabetic patients to suffer from kidney diseases. But Fig. 1 shows out of 364 DFI patients only 44 are having SRI. Also, HbA1c% of patients with SRI is lower than patients who do not have SRI, as indicated in Table 1. Both these data are contradicting to the statement made in line number 7.

2. Conclusion part should be rewritten for clarity (refer to line 257). In line 264, no need to write the values in bracket as it is already mentioned.

3. Some graphs/diagrams corresponding to the statistical data table should be great to visualize the important parameters.

4. A list of abbreviations will be helpful for the readers, as there are too many abbreviations used.

5. As the authors declare, "a prospective study with a priori calculated sample size is needed to verify the reliability of our conclusions" - it should be justified the relevance of study with the present data set with a future scope of research.

6. PLOS authors have the option to publish the peer review history of their article (what does this mean?). If published, this will include your full peer review and any attached files.

Reviewer #1: No

Reviewer #2: No

Reviewer #3: No

---

## [Author Response · Author response to Decision Letter 0]

1 Feb 2022

We have addressed the novelty and study concerns as mentioned by reviewers, and discussed in details about the excluded subjects.

---

## [Decision Letter · Decision Letter 1]

18 Feb 2022

PONE-D-21-37187R1Decreased accuracy of erythrocyte sedimentation rate in diagnosing osteomyelitis in diabetic foot infection patients with severe renal impairment: a retrospective cross-sectional studyPLOS ONE

Dear Dr. Que,

Thank you for submitting your manuscript to PLOS ONE. After careful consideration, we feel that it has merit but does not fully meet PLOS ONE’s publication criteria as it currently stands. Therefore, we invite you to submit a revised version of the manuscript that addresses the points raised during the review process.

We look forward to receiving your revised manuscript.

Kind regards,

Kanhaiya Singh, Ph.D

Academic Editor

PLOS ONE

Journal Requirements:

Additional Editor Comments (if provided):

Please address the comment raised by Reviewer 1.

Reviewers' comments:

Reviewer's Responses to Questions

**Comments to the Author**

1. If the authors have adequately addressed your comments raised in a previous round of review and you feel that this manuscript is now acceptable for publication, you may indicate that here to bypass the “Comments to the Author” section, enter your conflict of interest statement in the “Confidential to Editor” section, and submit your "Accept" recommendation.

Reviewer #1: All comments have been addressed

Reviewer #3: All comments have been addressed

2. Is the manuscript technically sound, and do the data support the conclusions?

Reviewer #1: Yes

Reviewer #3: Yes

3. Has the statistical analysis been performed appropriately and rigorously? 

Reviewer #1: N/A

Reviewer #3: Yes

4. Have the authors made all data underlying the findings in their manuscript fully available?

Reviewer #1: Yes

Reviewer #3: Yes

5. Is the manuscript presented in an intelligible fashion and written in standard English?

Reviewer #1: Yes

Reviewer #3: Yes

6. Review Comments to the Author

Reviewer #1: Authors have answered all the previous questions raised during the first review.

Minor issues that need to be addressed:

Legends: lines 234-239: the authors may add some interpretations of the graph along with the methods used

Figure 4: the font needs to be unbolded so the image is easier to interpret

Reviewer #3: (No Response)

7. PLOS authors have the option to publish the peer review history of their article (what does this mean?). If published, this will include your full peer review and any attached files.

Reviewer #1: No

Reviewer #3: No

---

## [Author Response · Author response to Decision Letter 1]

21 Feb 2022

We have added figure legend section and made the font unbolded in figures.

---

## [Decision Letter · Decision Letter 2]

8 Mar 2022

Decreased accuracy of erythrocyte sedimentation rate in diagnosing osteomyelitis in diabetic foot infection patients with severe renal impairment: a retrospective cross-sectional study

PONE-D-21-37187R2

Dear Dr. Que,

We’re pleased to inform you that your manuscript has been judged scientifically suitable for publication and will be formally accepted for publication once it meets all outstanding technical requirements.

Kind regards,

Kanhaiya Singh, Ph.D

Academic Editor

PLOS ONE

Additional Editor Comments (optional):

Reviewers' comments:

Reviewer's Responses to Questions

**Comments to the Author**

1. If the authors have adequately addressed your comments raised in a previous round of review and you feel that this manuscript is now acceptable for publication, you may indicate that here to bypass the “Comments to the Author” section, enter your conflict of interest statement in the “Confidential to Editor” section, and submit your "Accept" recommendation.

Reviewer #1: All comments have been addressed

2. Is the manuscript technically sound, and do the data support the conclusions?

Reviewer #1: Yes

3. Has the statistical analysis been performed appropriately and rigorously? 

Reviewer #1: N/A

4. Have the authors made all data underlying the findings in their manuscript fully available?

Reviewer #1: Yes

5. Is the manuscript presented in an intelligible fashion and written in standard English?

Reviewer #1: Yes

6. Review Comments to the Author

Reviewer #1: (No Response)

7. PLOS authors have the option to publish the peer review history of their article (what does this mean?). If published, this will include your full peer review and any attached files.

Reviewer #1: No

---

## [Editor Report · Acceptance letter]

12 Mar 2022

PONE-D-21-37187R2 

Decreased accuracy of erythrocyte sedimentation rate in diagnosing osteomyelitis in diabetic foot infection patients with severe renal impairment: a retrospective cross-sectional study 

Dear Dr. Que:

I'm pleased to inform you that your manuscript has been deemed suitable for publication in PLOS ONE. Congratulations! Your manuscript is now with our production department. 

Kind regards, 

on behalf of

Dr. Kanhaiya Singh 

Academic Editor

PLOS ONE